# Dual Policy Iteration

**Wen Sun**[1], **Geoffrey J. Gordon**[1]**, Byron Boots**[2]**, and J. Andrew Bagnell**[3]

[1]School of Computer Science, Carnegie Mellon University, USA
[2]College of Computing, Georgia Institute of Technology, USA
[3]Aurora Innovation, USA
{wensun, ggordon, dbagnell}@cs.cmu.edu, bboots@cc.gatech.edu

## Abstract

A novel class of Approximate Policy Iteration (API) algorithms have recently demonstrated impressive practical performance (*e.g.*, ExIt [1], AlphaGo-Zero [2]). This new family of algorithms maintains, and alternately optimizes, two policies: a fast, reactive policy (*e.g.*, a deep neural network) deployed at test time, and a slow, non-reactive policy (e.g., Tree Search), that can plan multiple steps ahead. The reactive policy is updated under supervision from the non-reactive policy, while the non-reactive policy is improved via guidance from the reactive policy. In this work we study this class of Dual Policy Iteration (DPI) strategy in an *alternating optimization framework* and provide a convergence analysis that extends existing API theory. We also develop a special instance of this framework which reduces the update of non-reactive policies to model-based optimal control using learned local models, and provides a theoretically sound way of unifying model-free and model-based RL approaches with unknown dynamics. We demonstrate the efficacy of our approach on various continuous control Markov Decision Processes.

## 1 Introduction

Approximate Policy Iteration (API) [3, 4, 5, 6, 7], including conservative API (CPI) [5], API driven by learned critics [8], or gradient-based API with stochastic policies [9, 10, 11, 12], have played a central role in Reinforcement Learning (RL) for decades and motivated many modern practical RL algorithms. Several existing API methods [4, 5] can provide both local optimality guarantees and global guarantees under strong assumptions regarding the way samples are generated (e.g., access to a reset distribution that is similar to the optimal policy's state distribution). However, most modern practical API algorithms rely on myopic random exploration (e.g., REINFORCE [13] type policy gradient or $\epsilon$-greedy). Sample inefficiency due to random exploration can cause even sophisticated RL methods to perform worse than simple black-box optimization with random search in parameter space [14].

Recently, a new class of API algorithms, which we call *Dual Policy Iteration* (DPI), has begun to emerge. These algorithms follow a richer strategy for improving the policy, with two policies under consideration at any time during training: a reactive policy, usually learned by some form of function approximation, used for generating samples and deployed at test time, and an intermediate policy that can only be constructed or accessed during training, used as an expert policy to guide the improvement of the reactive policy. For example, ExIt [1] maintains and updates a UCT-based policy [15] as an intermediate expert. ExIt then updates the reactive policy by directly imitating the tree-based policy which we expect would be *better* than the reactive policy as it involves a multi-step lookahead search. AlphaGo-Zero [2] employs a similar strategy to achieve super-human performance at the ancient game of Go. The key difference that distinguishes ExIt and AlphaGo-Zero from previous APIs is that they *leverage models to perform systematic forward search*: the policy resulting from forward search acts as an expert and directly informs the improvement direction for the reactive policy. Hence the

reactive policy improves by imitation instead of trial-and-error reinforcement learning. This strategy often provides better sample efficiency in practice compared to algorithms that simply rely on locally random search (e.g., AlphaGo-Zero abandons REINFORCE from AlphaGo [16]).

In this work we provide a general framework for synthesizing and analyzing DPI by considering a particular *alternating optimization strategy* with different optimization approaches each forming a new family of approximate policy iteration methods. We additionally consider the extension to the RL setting with *unknown dynamics*. For example, we construct a simple instance of our framework, where the intermediate expert is computed from *Model-Based Optimal Control* (MBOC) locally around the reactive policy, and the reactive policy in turn is updated incrementally under the guidance of MBOC. The resulting algorithm iteratively learns a local dynamics model, applies MBOC to compute a locally optimal policy, and then updates the reactive policy by imitation and achieve larger policy improvement per iteration than classic APIs. The instantiation shares similar spirit from some previous works from robotics and control literature, including works from [17, 18] and Guided Policy Search (GPS) [19] (and its variants (e.g., [20, 21, 22])), i.e., using local MBOC to speed up learning global policies.

To evaluate our approach, we demonstrate our algorithm on discrete MDPs and continuous control tasks. We show that by integrating local model-based search with learned local dynamics into policy improvement via an imitation learning-style update, our algorithm is substantially more sample-efficient than classic API algorithms such as CPI [5], as well as more recent actor-critic baselines [23], albeit at the cost of slower computation per iteration due to the model-based search. We also apply the framework to a *robust policy optimization* setting [24, 25] where the goal is to learn a *single* policy that can generalize across environments. In summary, the major practical difference between DPI and many modern practical RL approaches is that instead of relying on random exploration, the DPI framework integrates local model learning, local model-based search for advanced exploration, and an imitation learning-style policy improvement, to improve the policy in a more systematic way.

We also provide a general convergence analysis to support our empirical findings. Although our analysis is similar to CPI's, it has a key difference: as long as MBOC succeeds, we can provide a larger policy improvement than CPI at each iteration. Our analysis is general enough to provide theoretical intuition for previous successful practical DPI algorithms such as Expert Iteration (ExIt) [1]. We also analyze how predictive error from a learned local model can mildly affect policy improvement and show that locally accurate dynamics—a model that accurately predicts next states *under the current policy's state-action distribution*, is enough for improving the current policy. We believe our analysis of local model predictive error versus local policy improvement can shed light on further development of model-based RL approaches with learned local models. In summary, DPI operates in the middle of two extremes: (1) API type methods that update policies locally (e.g., first-order methods like policy gradient and CPI), (2) global model-based optimization where one attempts to learn a global model and perform model-based search. First-order methods have small policy improvement per iteration and learning a global model displays greater *model bias* and requires a dataset that covers the entire state space. DPI instead learns a local model and allows us to integrate models to leverage the power of model-based optimization to locally improve the reactive policy.

## 2  Preliminaries

A discounted infinite-horizon Markov Decision Process (MDP) is defined as $(\mathcal{S}, \mathcal{A}, P, c, \rho_0, \gamma)$. Here, $\mathcal{S}$ is a set of states, $\mathcal{A}$ is a set of actions, and $P$ is the transition dynamics: $P(s'|s, a)$ is the probability of transitioning to state $s'$ from state $s$ by taking action $a$. We use $P_{s,a}$ in short for $P(\cdot|s, a)$. We denote $c(s, a)$ as the cost of taking action $a$ while in state $s$. Finally, $\rho_0$ is the initial distribution of states, and $\gamma \in (0, 1)$ is the discount factor. Throughout this paper, we assume that we *know* the form of the cost function $c(s, a)$, but the transition dynamics $P$ are *unknown*. We define a stochastic policy $\pi$ such that for any state $s \in \mathcal{S}$, $\pi(\cdot|s)$ outputs a distribution over action space. The distribution of states at time step $t$, induced by running the policy $\pi$ until and including $t$, is defined $\forall s_t$: $d_\pi^t(s_t) = \sum_{\{s_i, a_i\}_{i \leq t-1}} \rho_0(s_0) \prod_{i=0}^{t-1} \pi(a_i|s_i) P(s_{i+1}|s_i, a_i)$, where by definition $d_\pi^0(s) = \rho_0(s)$ for any $\pi$. The state visitation distribution can be computed $d_\pi(s) = (1 - \gamma) \sum_{t=0}^{\infty} \gamma^t d_\pi^t(s)$. Denote $(d_\pi \pi)$ as the joint state-action distribution such that $d_\pi \pi(s, a) = d_\pi(s) \pi(a|s)$. We define the value

function $V^\pi(s)$, state-action value function $Q^\pi(s, a)$, and the objective function $J(\pi)$ as:

$$V^\pi(s) = \mathbb{E}\left[\sum_{t=0}^{\infty} \gamma^t c(s_t, a_t)|s_0 = s\right], Q^\pi(s, a) = c(s, a) + \gamma \mathbb{E}_{s' \sim P_{s,a}}[V^\pi(s')], J(\pi) = \mathbb{E}_{s \sim \rho_0}[V^\pi(s)].$$

With $V^\pi$ and $Q^\pi$, the advantage function $A^\pi(s, a)$ is defined as $A^\pi(s, a) = Q^\pi(s, a) - V^\pi(s)$. As we work in the cost setting, in the rest of the paper we refer to $A^\pi$ as the *disadvantage* function. The goal is to learn *a single stationary* policy $\pi^*$ that minimizes $J(\pi)$: $\pi^* = \arg\min_{\pi \in \Pi} J(\pi)$.

For two distributions $P_1$ and $P_2$, $D_{TV}(P_1, P_2)$ denotes *total variation distance*, which is related to the $L_1$ norm as $D_{TV}(P_1, P_2) = \|P_1 - P_2\|_1/2$ (if we have a finite probability space) and $D_{KL}(P_1, P_2) = \int_x P_1(x) \log(P_1(x)/P_2(x)) \mathrm{d}x$ denotes the KL divergence.

We introduce *Performance Difference lemma* (PDL) [5], which will be used extensively in this work:

**Lemma 2.1** *For any two policies $\pi$ and $\pi'$, we have:* $J(\pi) - J(\pi') = \frac{1}{1-\gamma}\mathbb{E}_{(s,a) \sim d_\pi \pi}\left[A^{\pi'}(s, a)\right].$

## 3 Dual Policy Iteration

We propose an alternating optimization framework inspired by the PDL (Lemma 2.1). Consider the min-max optimization framework: $\min_{\pi \in \Pi} \max_{\eta \in \Pi} \mathbb{E}_{s \sim d_\pi}\left[\mathbb{E}_{a \sim \pi(\cdot|s)}[A^\eta(s, a)]\right]$. It is not hard to see that the unique Nash equilibrium for the above equation is $(\pi, \eta) = (\pi^*, \pi^*)$. The above min-max proposes a general strategy, which we call Dual Policy Iteration (DPI): alternatively fix one policy and update the second policy. Mapping to previous practical DPI algorithms [1, 2], $\pi$ stands for the fast reactive policy and $\eta$ corresponds to the tree search policy. For notation purposes, we use $\pi_n$ and $\eta_n$ to represent the two policies in the $n^{\text{th}}$ iteration. Below we introduce one instance of DPI for settings with unknown models (hence no tree search), first describe how to compute $\eta_n$ from a given reactive policy $\pi_n$ (Sec. 3.1), and then describe how to update $\pi_n$ to $\pi_{n+1}$ via imitating $\eta_n$ (Sec. 3.2).

### 3.1 Updating $\eta$ with MBOC using Learned Local Models

Given $\pi_n$, the objective function for $\eta$ becomes: $\max_\eta \mathbb{E}_{s \sim d_{\pi_n}}\left[\mathbb{E}_{a \sim \pi_n(\cdot|s)}[A^\eta(s, a)]\right]$. From PDL we can see that updating $\eta$ is equivalent to finding the optimal policy $\pi^*$: $\arg\max_\eta(J(\pi_n) - J(\eta)) \equiv \arg\min_\eta J(\eta)$, regardless of what $\pi_n$ is. As directly minimizing $J(\eta)$ is as hard as the original problem, we update $\eta$ locally by constraining it to a trust region around $\pi_n$:

$$\arg\min_\eta J(\eta), \quad s.t., \mathbb{E}_{s \sim d_{\pi_n}} D_{TV}[(\eta(\cdot|s), \pi_n(\cdot|s))] \leq \alpha. \tag{1}$$

To solve the constraint optimization problem in Eq 1, we propose to learn $P_{s,a}$ and use it with any off-the-shelf model-based optimal control algorithm. Moreover, thanks to the trust region, we can simply learn a *local* dynamics model, *under the state-action distribution* $d_{\pi_n}\pi_n$. We denote the optimal solution to the above constrained optimization (Eq. 1) under the *real* model $P_{s,a}$ as $\eta_n^*$. Note that, due to the definition of the optimality, $\eta_n^*$ must perform better than $\pi_n$: $J(\pi_n) - J(\eta_n^*) \geq \Delta_n(\alpha)$, where $\Delta_n(\alpha) \geq 0$ is the performance gain from $\eta_n^*$ over $\pi_n$. When the trust region expands, i.e., $\alpha$ increases, then $\Delta_n(\alpha)$ approaches the performance difference between the optimal policy $\pi^*$ and $\pi_n$.

To perform MBOC, we learn a locally accurate model—a model $\hat{P}$ that is close to $P$ *under the state-action distribution induced by* $\pi_n$: we seek a model $\hat{P}$, such that the quantity $\mathbb{E}_{(s,a) \sim d_{\pi_n}\pi_n} D_{TV}(\hat{P}_{s,a}, P_{s,a})$ is small. Optimizing $D_{TV}$ directly is hard, but note that, by Pinsker's inequality, we have $D_{KL}(P_{s,a}, \hat{P}_{s,a}) \geq D_{TV}(\hat{P}_{s,a}, P_{s,a})^2$, which indicates that we can optimize a surrogate loss defined by a KL-divergence:

$$\arg\min_{\hat{P} \in \mathbf{P}} \mathbb{E}_{s \sim d_{\pi_n}, a \sim \pi_n(s)} D_{KL}(P_{s,a}, \hat{P}_{s,a}) = \arg\min_{\hat{P} \in \mathbf{P}} \mathbb{E}_{s \sim d_{\pi_n}, a \sim \pi_n(s), s' \sim P_{s,a}}[-\log \hat{P}_{s,a}(s')], \tag{2}$$

where we denote $\mathbf{P}$ as the model class. Hence we reduce the local model fitting problem into a classic maximum likelihood estimation (MLE) problem, where the training data $\{s, a, s'\}$ can be easily collected by executing $\pi_n$ on the real system (i.e., $P_{s,a}$). As we will show later, to ensure policy improvement, we just need a learned model to perform well under $d_{\pi_n}\pi_n$ (i.e., no training and

testing distribution mismatch as one will have for global model learning). For later analysis purposes, we denote $\hat{P}$ as the MLE in Eq. 2 and assume $\hat{P}$ is $\delta$-optimal under $d_{\pi_n}\pi_n$:

$$\mathbb{E}_{(s,a)\sim d_{\pi_n}\pi_n} D_{TV}(\hat{P}_{s,a}, P_{s,a}) \leq \delta, \tag{3}$$

where $\delta \in \mathbb{R}^+$ is controlled by the complexity of model class $\mathbf{P}$ and by the amount of training data we sample using $\pi_n$, which can be analyzed by standard supervised learning theory. After achieving a locally accurate model $\hat{P}$, we solve Eq. 1 using any existing stochastic MBOC solvers. Assume a MBOC solver returns an optimal policy $\eta_n$ under the estimated model $\hat{P}$ subject to trust-region:

$$\eta_n = \arg\min_\pi J(\pi), s.t., \ s_{t+1} \sim \hat{P}_{s_t, a_t}, \ \mathbb{E}_{s\sim d_{\pi_n}} D_{TV}(\pi, \pi_n) \leq \alpha. \tag{4}$$

At this point, a natural question is: If $\eta_n$ is solved by an MBOC solver under $\hat{P}$, by how much can $\eta_n$ outperform $\pi_n$ when *executed under the real dynamics* $P$? Recall that the performance gap between the real optimal solution $\eta_n^*$ (optimal under $P$) and $\pi_n$ is denoted as $\Delta_n(\alpha)$. The following theorem quantifies the performance gap between $\eta_n$ and $\pi_n$ using the learned local model's predictive error $\delta$:

**Theorem 3.1** *Assume $\hat{P}_{s,a}$ satisfies Eq. 3, and $\eta_n$ is the output of a MBOC solver for the optimization problem defined in Eq. 4, then we have:*

$$J(\eta_n) \leq J(\pi_n) - \Delta_n(\alpha) + O\left(\frac{\gamma\delta}{1-\gamma} + \frac{\gamma\alpha}{(1-\gamma)^2}\right).$$

The proof of the above theorem can be found in Appendix A.2. Theorem 3.1 indicates that when the model is *locally accurate*, i.e., $\delta$ is small (e.g., $\mathbf{P}$ is rich and we have enough data from $d_{\pi_n}\pi_n$), $\alpha$ is small, and there exists a local optimal solution that is significantly better than the current policy $\pi_n$ (i.e., $\Delta_n(\alpha) \in \mathbb{R}^+$ is large), then the OC solver with the learned model $\hat{P}$ finds a nearly local-optimal solution $\eta_n$ that outperforms $\pi_n$. With a better $\eta_n$, now we are ready to improve $\pi_n$ via imitating $\eta_n$.

### 3.2 Updating $\pi$ via Imitating $\eta$

Given $\eta_n$, we compute $\pi_{n+1}$ by performing the following constrained optimization procedure:

$$\arg\min_\pi \mathbb{E}_{s\sim d_{\pi_n}} \left[\mathbb{E}_{a\sim\pi(\cdot|s)} \left[A^{\eta_n}(s,a)\right]\right], s.t., \mathbb{E}_{s\sim d_{\pi_n}} \left[D_{TV}(\pi(\cdot|s), \pi_n(\cdot|s))\right] \leq \beta \tag{5}$$

Note that the key difference between Eq. 5 and classic API policy improvement procedure is that we use $\eta_n$'s disadvantage function $A^{\eta_n}$, i.e., we are performing imitation learning by treating $\eta_n$ as an expert in this iteration [26, 27]. We can solve Eq. 5 by converting it to supervised learning problem such as cost-sensitive classification [5] by sampling states and actions from $\pi_n$ and estimating $A^{\eta_n}$ via rolling out $\eta_n$, subject to an L1 constraint.

Note that a CPI-like update approximately solves the above constrained problem as well:

$$\pi_{n+1} = (1-\beta)\pi_n + \beta\pi_n^*, \text{ where } \pi_n^* = \arg\min_\pi \mathbb{E}_{s\sim d_{\pi_n}} \left[\mathbb{E}_{a\sim\pi(\cdot|s)}[A^{\eta_n}(s,a)]\right]. \tag{6}$$

Note that $\pi_{n+1}$ satisfies the constraint as $D_{TV}(\pi_{n+1}(\cdot|s), \pi_n(\cdot|s)) \leq \beta, \forall s$. Intuitively, the update in Eq. 6 can be understood as first solving the objective function to obtain $\pi_n^*$ without considering the constraint, and then moving $\pi_n$ towards $\pi_n^*$ until the boundary of the constraint is reached.

### 3.3 DPI: Combining Updates on $\pi$ and $\eta$

In summary, assume MBOC is used for Eq. 1, DPI operates in an iterative way: with $\pi_n$:

1. Fit MLE $\hat{P}$ on states and actions from $d_{\pi_n}\pi_n$ (Eq. 2).
2. $\eta_n \leftarrow$ MBOC($\hat{P}$), subject to trust region $\mathbb{E}_{s\sim d_{\pi_n}} D_{TV}(\pi, \pi_n) \leq \alpha$ (Eq. 4)
3. Update to $\pi_{n+1}$ by imitating $\eta_n$, subject to trust region $\mathbb{E}_{s\sim d_{\pi_n}} D_{TV}(\pi, \pi_n) \leq \beta$ (Eq. 5).

The above framework shows how $\pi$ and $\eta$ are tightened together to guide each other's improvements: the first step corresponds classic MLE under $\pi_n$'s state-action distribution: $d_{\pi_n}\pi_n$; the second step corresponds to model-based policy search around $\pi_n$ ($\hat{P}$ is only locally accurate); the third step corresponds to updating $\pi$ by imitating $\eta$ (i.e., imitation). Note that in practice MBOC solver (e.g., a second order optimization method, as we will show in our practical algorithm below) could be computationally expensive and slow (e.g. tree search in ExIt and AlphaGo-Zero), but once $\hat{P}$ is provided, MBOC does not require additional samples from the real system.

**Connections to Previous works** We can see that the above framework generalizes several previous work from API and IL. **(a)** If we set $\alpha = 0$ in the limit, we reveal CPI (assuming we optimize with Eq. 6), i.e., no attempt to search for a better policy using model-based optimization. **(b)** Mapping to ExIt, our $\eta_n$ plays the role of the tree-based policy, and our $\pi_n$ plays the role of the apprentice policy, and MBOC plays the role of forward search. **(c)** when an optimal expert policy $\pi^*$ is available during and only during training, we can set every $\eta_n$ to be $\pi^*$, and DPI then reveals a previous IL algorithm—AGGREVATED [27].

## 4   Analysis of Policy Improvement

We provide a general convergence analysis for DPI. The trust region constraints in Eq. 1 and Eq. 5 tightly combines MBOC and policy improvement together, and is the key to ensure monotonic improvement and achieve larger policy improvement per iteration than existing APIs.

Define $\mathbb{A}_n(\pi_{n+1})$ as the disadvantage of $\pi^{n+1}$ over $\eta_n$ under $d_{\pi_n}$: $\mathbb{A}_n(\pi_{n+1}) = \mathbb{E}_{s\sim d_{\pi_n}} \left[ \mathbb{E}_{a\sim\pi_{n+1}(\cdot|s)} \left[ A^{\eta_n}(s,a) \right] \right]$. Note that $\mathbb{A}_n(\pi_{n+1})$ is at least non-positive (if $\pi$ and $\eta$ are from the same function class, or $\pi$'s policy class is rich enough to include $\eta$), as if we set $\pi_{n+1}$ to $\eta_n$. In that case, we simply have $\mathbb{A}_n(\pi_{n+1}) = 0$, which means we can hope that the IL procedure (Eq. 5) finds a policy $\pi_{n+1}$ that achieves $\mathbb{A}_n(\pi_{n+1}) < 0$ (i.e., local improvement over $\eta_n$). The question we want to answer is: by *how much* is the performance of $\pi_{n+1}$ improved over $\pi_n$ by solving the two trust-region optimization procedures detailed in Eq. 1 and Eq. 5. Following Theorem 4.1 from [5], we define $\varepsilon = \max_s |\mathbb{E}_{a\sim\pi_{n+1}(\cdot|s)}[A^{\eta_n}(s,a)]|$, which measures the maximum possible one-step improvement one can achieve from $\eta_n$. The following theorem states the performance improvement:

**Theorem 4.1** *Solve Eq. 1 to get $\eta_n$ and Eq. 5 to get $\pi_{n+1}$. The improvement of $\pi_{n+1}$ over $\pi_n$ is:*

$$J(\pi_{n+1}) - J(\pi_n) \leq \frac{\beta\varepsilon}{(1-\gamma)^2} - \frac{|\mathbb{A}_n(\pi_{n+1})|}{1-\gamma} - \Delta_n(\alpha). \tag{7}$$

The proof of Theorem 4.1 is provided in Appendix A.3. When $\beta$ is small, we are guaranteed to find a policy $\pi_{n+1}$ where the total cost decreases by $\Delta_n(\alpha) + |\mathbb{A}_n(\pi_{n+1})|/(1-\gamma)$ compared to $\pi_n$. Note that classic CPI's per iteration improvement [5, 12] only contains a term that has the similar meaning and magnitude of the second term in the RHS of Eq. 7. Hence DPI can improve the performance of CPI by introducing an extra term $\Delta_n(\alpha)$, and the improvement could be substantial when there exists a locally optimal policy $\eta_n$ that is much better than the current reactive policy $\pi_n$. Such $\Delta(\alpha)$ comes from the explicit introduction of a model-based search into the training loop, which does not exist in classic APIs. From a practical point view, modern MBOCs are usually second-order methods, while APIs are usually first-order (e.g., REINFORCE and CPI). Hence it is reasonable to expect $\Delta(\alpha)$ itself will be larger than API's policy improvement per iteration. Connecting back to ExIt and AlphaGo-Zero under model-based setting, $\Delta(\alpha)$ stands for the improvement of the tree-based policy over the current deep net reactive policy. In ExIt and AlphaGo Zero, the tree-based policy $\eta_n$ performs fixed depth forward search followed by rolling out $\pi_n$ (i.e., bottom up by $V^{\pi_n}(s)$), which ensures the expert $\eta_n$ outperforms $\pi_n$.

When $|\Delta_n(\alpha)|$ and $|\mathbb{A}_n(\pi_{n+1})|$ are small, i.e., $|\Delta_n(\alpha)| \leq \xi$ and $|\mathbb{A}_n(\pi_{n+1})| \leq \xi$, then we can guarantee that $\eta_n$ and $\pi_n$ are good policies, *under the stronger assumption that the initial distribution $\rho_0$ happens to be a good distribution (e.g., close to $d_{\pi^*}$), and the realizable assumption*: $\min_{\pi\in\Pi} \mathbb{E}_{s\sim d_{\pi_n}} \left[ \mathbb{E}_{a\sim\pi(\cdot|s)}[A^{\eta_n}(s,a)] \right] = \mathbb{E}_{s\sim d_{\pi_n}} [\min_{a\in\mathcal{A}} [A^{\eta_n}(s,a)]]$, holds. We show in Appendix A.4 that under the realizable assumption:

$$J(\eta_n) - J(\pi^*) \leq \left( \max_s \left( \frac{d_{\pi^*}(s)}{\rho_0(s)} \right) \right) \left( \frac{\xi}{\beta(1-\gamma)^2} + \frac{\xi}{\beta(1-\gamma)} \right).$$

The term $(\max_s (d_{\pi^*}(s)/\rho_0(s)))$ measures the distribution mismatch between the initial distribution $\rho_0$ and the optimal policy $\pi^*$, and appears in some previous API algorithms–CPI [5] and PSDP [4]. A $\rho_0$ that is closer to $d_{\pi^*}$ (e.g., let experts reset the agent's initial position if possible) ensures better global performance guarantee. CPI considers a setting where a good reset distribution $\nu$ (different from $\rho_0$) is available, DPI can leverage such reset distribution by replacing $\rho_0$ by $\nu$ at training.

In summary, we can expect larger per-iteration policy improvement from DPI compared to CPI (and TRPO which has similar per iteration policy improvement as CPI), thanks to the introduction of local model-based search. The final performance bound of the learned policy is in par with CPI and PSDP.

# 5   An Instance of DPI

In this section, we dive into the details of each update step of DPI and suggest one practical instance of DPI, which can be used in continuous control settings. We denote $T$ as the maximum possible horizon.[1] We denote the state space $\mathcal{S} \subseteq \mathbb{R}^{d_s}$ and action space $\mathcal{A} \subseteq \mathbb{R}^{d_a}$. We work on parameterized policies: we parameterize policy $\pi$ as $\pi(\cdot|s; \theta)$ for any $s \in \mathcal{S}$ (e.g., a neural network with parameter $\theta$), and parameterize $\eta$ by a sequence of time-varying linear-Gaussian policies $\eta = \{\eta_t\}_{1 \leq t \leq T}$, where $\eta_t(\cdot|s) = \mathcal{N}(K_t s + k_t, P_t)$ with control gain $K_t \in \mathbb{R}^{d_a \times d_s}$, bias term $k_t \in \mathbb{R}^{d_a}$ and Covariance $P_t \in \mathbb{R}^{d_a \times d_a}$. We will use $\Theta = \{K_t, k_t, P_t\}_{0 \leq t \leq H}$ to represent the collection of the parameters of all the linear-Gaussian policies across the entire horizon. One approximation we make here is to replace the policy divergence measure $D_{TV}(\pi_n, \pi)$ (note total variation distance is symmetric) with the KL-divergence $D_{KL}(\pi_n, \pi)$, which allows us to leverage Natural Gradient [11, 10].[2] To summarize, $\pi_n$ and $\eta_n$ are short for $\pi_{\theta_n}$ and $\eta_{\Theta_n} = \{\mathcal{N}(K_t s + k_t, P_t)\}_t$, respectively. Below we first describe how to compute $\eta_{\Theta_n}$ given $\pi_n$ (Sec. 5.1), and then describe how to update $\pi$ via imitating $\eta_{\Theta_n}$ using Natural Gradient (Sec. 5.2).

## 5.1   Updating $\eta_\Theta$ with MBOC using Learned Time Varying Linear Models

We explain here how to find $\eta_n$ given $\pi_n$ using MBOC. In our implementation, we use Linear Quadratic Gaussian (LQG) optimal control [28] as the black-box optimal control solver. We learn a sequence of time varying linear Gaussian transition models to represent $\hat{P}$: $\forall t \in [1, T]$,

$$s_{t+1} \sim \mathcal{N}(A_t s_t + B_t a_t + c_t, \Sigma_t), \quad (8)$$

where $A_t, B_t, c_t, \Sigma_t$ can be learned using classic linear regression techniques on a dataset $\{s_t, a_t, s_{t+1}\}$ collected from executing $\pi_n$ on the real system. Although the dynamics $P(s, a)$ may be complicated over the entire space, linear dynamics could locally approximate the dynamics well (after all, our theorem only requires $\hat{P}$ to have low predictive error under $d_{\pi_n}\pi_n$).

---

**Algorithm 1** AGGREVATED-GPS

1: **Input:** Parameters $\alpha \in \mathbb{R}^+$, $\beta \in \mathbb{R}^+$.
2: Initialized $\pi_{\theta_0}$
3: **for** n = 0 to ... **do**
4:      Execute $\pi_{\theta_n}$ to generate a set of trajectories
5:      Fit local linear dynamics $\hat{P}$ (Eq. 8) using $\{s_t, a_t, s_{t+1}\}$ collected from step 1
6:      Solve the minmax in Eq. 9 subject to $\hat{P}$ to obtain $\eta_{\Theta_n}$ and form disadvantage $A^{\eta_{\Theta_n}}$
7:      Compute $\theta_{n+1}$ by NGD (Eq. 12)
8: **end for**

---

Next, to find a locally optimal policy under linear-Gaussian transitions (i.e., Eq. 4), we add the KL constraint to the objective with Lagrange multiplier $\mu$ and form an equivalent min-max problem:

$$\min_\eta \max_{\mu \geq 0} \mathbb{E}\left[\sum_{t=1}^T \gamma^{t-1} c(s_t, a_t)\right] + \mu\left(\sum_{t=1}^T \gamma^{t-1} \mathbb{E}_{s \sim d_\eta^t}[D_{KL}(\eta, \pi_n)] - \alpha\right), \quad (9)$$

where $\mu$ is the Lagrange multiplier, which can be solved by alternatively updating $\eta$ and $\mu$ [19]. For a fixed $\mu$, using the derivation from [19], ignoring terms that do not depend on $\eta$, Eq. 9 can be written:

$$\arg\min_\eta \mathbb{E}\left[\sum_{t=1}^T \gamma^{t-1}(c(s_t, a_t)/\mu - \log \pi_n(a_t|s_t))\right] - \sum_{t=1}^T \gamma^{t-1} \mathbb{E}_{s \sim d_\eta^t}[\mathcal{H}(\eta(\cdot|s))], \quad (10)$$

where $\mathcal{H}(\pi(\cdot|s)) = \sum_a \pi(a|s) \ln(\pi(a|s))$ is the negative entropy. Hence the above formulation can be understood as using a *new cost function*: $c'(s_t, a_t) = c(s_t, a_t)/\mu - \log(\pi_n(a_t|s_t))$, and an entropy regularization on $\pi$. It is well known in the optimal control literature that when $c'$ is quadratic and dynamics are linear, the optimal sequence of linear Gaussian policies for the objective in Eq. 10 can be found exactly by a Dynamic Programming (DP) based approach, the *Linear Quadratic Regulator* (LQR) [28]. Given a dataset $\{(s_t, a_t), c'(s_t, a_t)\}$ collected while executing $\pi_n$, we can fit a quadratic approximation of $c'(s, a)$ [29, 19]. With a quadratic approximation of $c'$ and linear dynamics, we solve Eq. 10 for $\eta$ exactly by LQR [29]. Once we get $\eta$, we go back to Eq. 9 and update the Lagrange multiplier $\mu$, for example, by projected gradient ascent [30]. Upon convergence, LQR gives us a sequence of time-dependent linear Gaussian policies together with a sequence of analytic quadratic cost-to-go functions $Q_t(s, a)$, and quadratic disadvantage functions $A_t^{\eta_{\Theta_n}}(s, a)$, for all $t \in [T]$.

## 5.2 Updating $\pi_\theta$ via imitating $\eta_\Theta$ using Natural Gradient

Performing a second order Taylor expansion of the KL constraint $\mathbb{E}_{s \sim d_{\pi_n}}[D_{KL}(\pi_n(\cdot|a), \pi(\cdot|s;\theta)))]$ around $\theta_n$ [11, 10], we get the following constrained optimization problem:

$$\min_\theta \mathbb{E}_{s \sim d_{\pi_{\theta_n}}}[\mathbb{E}_{a \sim \pi(\cdot|s;\theta)}[A^{\eta_{\Theta_n}}(s,a)]], s.t., (\theta - \theta_n)^T F_{\theta_n}(\theta - \theta_n) \le \beta, \qquad (11)$$

where $F_{\theta_n}$ is the Hessian of the KL constraint $\mathbb{E}_{s \sim d_{\pi_{\theta_n}}} D_{KL}(\pi_{\theta_n}, \pi_\theta)$ (i.e., Fisher information matrix), measured at $\theta_n$. Denote the objective (i.e., the first term in Eq. 11) as $L_n(\theta)$, and denote $\nabla_{\theta_n}$ as $\nabla_\theta L_n(\theta)|_{\theta = \theta_n}$, we can optimize $\theta$ by performing natural gradient descent (NGD):

$$\theta_{n+1} = \theta_n - \mu F_{\theta_n}^{-1}\nabla_{\theta_n}, \text{where } \mu = \sqrt{\beta/(\nabla_{\theta_n}^T F_{\theta_n}^{-1}\nabla_{\theta_n})}. \qquad (12)$$

The specific $\mu$ above ensures the KL constraint is satisfied. More details about the imitation update on $\pi$ can be found in Appendix B.3.

**Summary**  If we consider $\eta$ as an expert, NGD is similar to natural gradient AGGREVATED—a differential IL approach [27]. We summarizes the procedures presented in Sec. 3.1&5.2 in Alg. 1, which we name as AGGREVATED-GPS, stands for the fact that we are using MBOC to Guide Policy Search [19, 21] via AGGREVATED-type update. Every iteration, we run $\pi_{\theta_n}$ on $P$ to gather samples. We estimate time dependent local linear dynamics $\hat{P}$ and then leverage an OC solver (e.g, LQR) to solve the Lagrangian in Eq. 9 to compute $\eta_{\Theta_n}$ and $A^{\eta_{\Theta_n}}$. We then perform NGD to update to $\pi_{n+1}$.

## 5.3 Additional Related Works

The most closely related work with respect to Alg. 1 is Guided Policy Search (GPS) for unknown dynamics [19] and its variants (e.g.,[20, 21, 22]). GPS (including its variants) demonstrates model-based optimal control approaches can be used to speed up training policies parameterized by rich non-linear function approximators (e.g., deep networks) in large-scale applications. While Alg. 1 in high level follows GPS's iterative procedure of alternating reactive policy improvement and MBOC, the main difference between Alg. 1 and GPSs are the update procedure of the reactive policy. Classic GPS, including the mirror descent version, phrases the update procedure of the reactive policy as a *behavior cloning procedure*, i.e., given an expert policy $\eta$, we perform $\min_\pi D_{KL}(d_\mu \mu || d_\pi \pi)$ [3]. Note that our approach to updating $\pi$ is fundamentally on-policy, *i.e.,* we generate samples from $\pi$. Moreover, we update $\pi$ by performing policy iteration against $\eta$, i.e., $\pi$ approximately acts greedily with respect to $A^\eta$, which resulting a key difference: if we limit the power of MBOC, *i.e.*, set the trust region size in MBOC step to zero in both DPI and GPS, then our approach reduces to CPI and thus improves $\pi$ to local optimality. GPS and its variants, by contrast, have no ability to improve the reactive policy in that setting.

# 6 Experiments

We tested our approach on several MDPs: (1) a set of random discrete MDPs (Garnet problems [7]) (2) Cartpole balancing [31], (3) Helicopter Aerobatics (Hover and Funnel) [32], (4) Swimmer, Hopper and Half-Cheetah from the MuJoCo physics simulator [33]. The goals of these experiments are: **(a)** to experimentally verify that using $A^\eta$ from the intermediate expert $\eta$ computed by model-based search to perform policy improvement is more sample-efficient than using $A^\pi$. **(b)** to show that our approach can be applied to robust policy search and can outperform existing approaches [25].

## 6.1 Comparison to CPI on Discrete MDPs

Following [7], we randomly create ten discrete MDPs with 1000 states and 5 actions. Different from the techniques we introduced in Sec. 5.2 for continuous settings, here we use the conservative

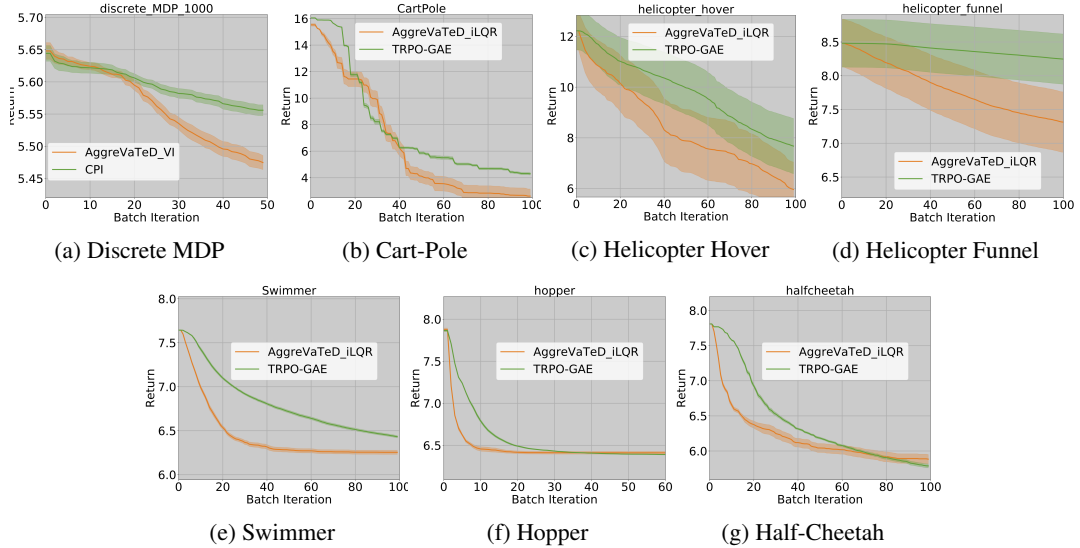

| (a) Discrete MDP | (b) Cart-Pole | (c) Helicopter Hover | (d) Helicopter Funnel |

| (e) Swimmer | (f) Hopper | (g) Half-Cheetah |

Figure 1: Performance (mean and standard error of cumulative cost in $\log_2$-scale on y-axis) versus number of episodes ($n$ on x-axis).

update shown in Eq. 6 to update the reactive policy, where each $\pi_n^*$ is a linear classifier and is trained using regression-based cost-sensitive classification on samples from $d_{\pi_n}$ [5]. The feature for each state $\phi(s)$ is a binary encoding of the state ($\phi(s) \in \mathbb{R}^{\log_2(|\mathcal{S}|)}$). We maintain the estimated transition $\hat{P}$ in a tabular representation. The policy $\eta$ is also in a tabular representation (hence expert $\eta$ and reactive policy $\pi$ have different feature representation) and is computed using exact VI under $\hat{P}$ and $c'(s, a)$ (hence we name our approach here as AGGREVATED-VI). The setup and the conservative update implementation is detailed in Appendix B.1. Fig. 1a reports the statistical performance of our approach and CPI over the 10 random discrete MDPs. Note that our approach is more sample-efficient than CPI, although we observed it is slower than CPI per iteration as we ran VI using learned model. We tune $\beta$ and neither CPI nor our approach uses line search on $\beta$. The major difference between AGGREVATED-VI and CPI here is that we used $A^\eta$ instead of $A^\pi$ to update $\pi$.

## 6.2 Comparison to Actor-Critic in Continuous Settings

We compare against TRPO-GAE [23] on a set of continuous control tasks. The setup is detailed in Appendix B.4. TRPO-GAE is a actor-critic-like approach where both actor and critic are updated using trust region optimization. We use a two-layer neural network to represent policy $\pi$ which is updated by natural gradient descent. We use LQR as the underlying MBOC solver and we name our approach as AGGREVATED-ILQR. Fig. 1 (b-g) shows the comparison between our method and TRPO-GAE over a set of continuous control tasks (confidence interval is computed from 20 random trials). As we can see, our method is significantly more sample-efficient than TRPO-GAE albeit slower per iteration as we perform MBOC. The major difference between our approach and TRPO-GAE is that we use $A^\eta$ while TRPO-GAE uses $A^\pi$ for the policy update. Note that both $A^\eta$ and $A^\pi$ are computed using the rollouts from $\pi$. The difference is that our approach uses rollouts to learn local dynamics and analytically estimates $A^\eta$ using MBOC, while TRPO-GAE learns $A^\pi$ using rollouts. Overall, our approach converges faster than TRPO-GAE (*i.e.*, uses less samples), which again indicates the benefit of using $A^\eta$ in policy improvement.

## 6.3 Application on Robust Policy Optimization

One application for our approach is robust policy optimization [34], where we have multiple training environments that are all potentially different from, but similar to, the testing environments. The goal is to train a *single* reactive policy using the training environments and deploy the policy on a test environment *without any further training*. Previous work suggests a policy that optimizes all the training models simultaneously is stable and robust during testing [24, 25], as the training environments together act as "regularization" to avoid overfitting and provide generalization.

More formally, let us assume that we have $M$ training environments. At iteration $n$ with $\pi_{\theta_n}$, we execute $\pi_{\theta_n}$ on the $i$'th environment, generate samples, fit local models, and call MBOC associated with the $i$'th environment to compute $\eta_{\Theta_n^i}$, for all $i \in [M]$. With $A^{\eta_{\Theta_n^i}}$, for all $i \in [M]$, we consider all training environments equally and formalize the objective $L_n(\theta)$ as $L_n(\theta) = \sum_{i=1}^{M} \mathbb{E}_{s \sim d_{\pi_{\theta_n}}} [\mathbb{E}_{a \sim \pi(\cdot|s;\theta)}[A^{\eta_{\Theta_n^i}}]]$. We update $\theta_n$ to $\theta_{n+1}$ by NGD on $L_n(\theta)$. Intuitively, we update $\pi_\theta$ by imitating $\eta_{\Theta_n^i}$ simultaneously for all $i \in [M]$.

We consider two simulation tasks, cartpole balancing and helicopter funnel. For each task, we create ten environments by varying the physical parameters (e.g., mass of helicopter, mass and length of pole). We use 7 of the environments for training and the remaining three for testing. We compare our algorithm against TRPO, which could be regarded as a model-free, natural gradient version of the first-order algorithm proposed in [25]. We also ran our algorithm on a single randomly picked training environment, but still tested on test environments, which is denoted as *non-robust* in Fig. 2. Fig. 2 summarizes the comparison

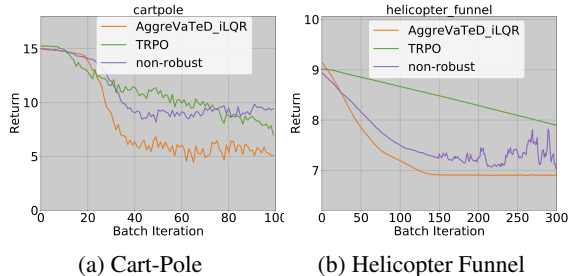

(a) Cart-Pole      (b) Helicopter Funnel

Figure 2: Performance (mean in log-scale on y-axis) versus episodes ($n$ on x-axis) in robust control.

between our approach and baselines. Similar to the trend we saw in the previous section, our approach is more sample-efficient in the robust policy optimization setup as well. It is interesting to see the "non-robust" approach fails to further converge, which illustrates the overfitting phenomenon: the learned policy overfits to one particular training environment.

# 7    Conclusion

We present and analyze Dual Policy Iteration—a framework that alternatively computes a non-reactive policy via more advanced and systematic search, and updates a reactive policy via imitating the non-reactive one. Recent algorithms that have been successful in practice, like AlphaGo-Zero and ExIt, are subsumed by the DPI framework. We then provide a simple instance of DPI for RL with unknown dynamics, where the instance integrates local model fit, local model-based search, and reactive policy improvement via imitating the teacher–the nearly local-optimal policy resulting from model-based search. We theoretically show that integrating model-based search and imitation into policy improvement could result in larger policy improvement at each step. We also experimentally demonstrate the improved sample efficiency compared to strong baselines.

Our work also opens some new problems. In theory, the performance improvement during one call of optimal control with the local accurate model depends on a term that scales quadratically with respect to the horizon $1/(1 - \gamma)$. We believe the dependency on horizon can be brought down by leveraging system identification methods focusing on multi-step prediction [35, 36]. On the practical side, our specific implementation has some limitations due to the choice of LQG as the underlying OC algorithm. LQG-based methods usually require the dynamics and cost functions to be somewhat smooth so that they can be locally approximated by polynomials. We also found that LQG planning horizons must be relatively short, as the approximation error from polynomials will likely compound over the horizon. We plan to explore the possibility of learning a non-linear dynamics and using more advanced non-linear optimal control techniques such as Model Predictive Control (MPC) for more sophisticated control tasks.

# Acknowledgement

We thank Sergey Levine for valuable discussion. WS is supported in part by Office of Naval Research contract N000141512365

## Footnotes

[1]Note $T$ is the maximum possible horizon which could be long. Hence, we still want to output a single policy, especially when the policy is parameterized by complicated non-linear function approximators like deep nets.

[2]Small $D_{KL}$ leads to small $D_{TV}$, as by Pinsker's inequality, $D_{KL}(q, p)$ (and $D_{KL}(p, q)) \geq D_{TV}(p, q)^2$.

[3]See Line 3 in Alg.2 in [21], where in principle a behavior cloning on $\pi$ uses samples from expert $\eta$ (i.e., off-policy samples). We note, however, in actual implementation some variants of GPS tend to swap the order of $\pi$ and $\eta$ inside the KL, often resulting a on-policy sampling strategy (e.g.,[22]). We also note a Mirror Descent interpretation and analysis to explain GPS's convergence [21] implies the correct way to perform a projection is to minimize the reverse KL, *i.e.*, $\arg\min_{\pi \in \Pi} D_{KL}(d_\pi \pi || d_\eta \eta)$. This in turn matches the DPI intuition: one should attempt to find a policy $\pi$ that is similar to $\eta$ under the state distribution of $\pi$ itself.

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
