[Supplementary Material]

# A  Missing Proofs

## A.1  Useful Lemmas

As we work in finite probability space, we will use the following fact regarding total variation distance and L1 distance for any two probability measures $P$ and $Q$:

$$\|P - Q\|_1 = 2D_{TV}(P, Q). \tag{13}$$

Recall that $d_\pi = (1 - \gamma) \sum_{t=0}^\infty \gamma^t d_\pi^t$. The following lemma shows that if two policies are close with each other in terms of the trust region constraint we defined in the paper, then the state visitations of the two policies are not that far away.

**Lemma A.1** *Given any two policy $\pi_1$ and $\pi_2$ such that $\mathbb{E}_{s \sim d_{\pi_1}} [D_{TV}(\pi_1(\cdot|s), \pi_2(\cdot|s))] \leq \alpha$, then we have:*

$$\|d_{\pi_1} - d_{\pi_2}\|_1 \leq \frac{2\alpha}{1 - \gamma}. \tag{14}$$

**Proof:**  Fix a state $s$ and time step $t$, let us first consider $d_{\pi_1}^t(s) - d_{\pi_2}^t(s)$.

$$
\begin{aligned}
&d_{\pi_1}^t(s) - d_{\pi_2}^t(s) \\
&= \sum_{s_0, s_1, \ldots, s_{t-1}} \sum_{a_0, a_1, \ldots, a_{t-1}} \Big( \rho(s_0) \pi_1(a_0|s_0) P_{s_0, a_0}(s_2) \ldots \pi_1(a_{t-1}|s_{t-1}) P_{s_{t-1}, a_{t-1}}(s) \\
&\quad - \rho(s_0) \pi_2(a_0|s_0) P_{s_0, a_0}(s_1) \ldots \pi_2(a_{t-1}|s_{t-1}) P_{s_{t-1}, a_{t-1}}(s) \Big) \\
&= \sum_{s_0} \rho(s_0) \sum_{a_0} \pi_1(a_0|s_0) \sum_{s_1} P_{s_0, a_0}(s_1) \ldots \sum_{a_{t-1}} \pi_1(a_{t-1}|s_{t-1}) P_{s_{t-1}, a_{t-1}}(s) \\
&\quad - \sum_{s_0} \rho(s_0) \sum_{a_0} \pi_2(a_0|s_0) \sum_{s_1} P_{s_0, a_0}(s_1) \ldots \sum_{a_{t-1}} \pi_2(a_{t-1}|s_{t-1}) P_{s_{t-1}, a_{t-1}}(s) \\
&= \sum_{s_0} \rho(s_0) \sum_{a_0} \pi_1(a_0|s_0) P(s_t = s|s_0, a_0; \pi_1) - \sum_{s_0} \rho(s_0) \sum_{a_0} \pi_2(a_0|s_0) P(s_t = s|s_0, a_0; \pi_2),
\end{aligned}
\tag{15}
$$

where $P(s_t = s|s_0, a_0; \pi)$ stands for the probability of reaching state $s$ at time step $t$, starting at $s_0$ and $a_0$ and then following $\pi$. Continue, we have:

$$
\begin{aligned}
&|d_{\pi_1}^t(s) - d_{\pi_2}^t(s)| \\
&= |\sum_{s_0} \rho(s_0) \sum_{a_0} \pi_1(a_0|s_0) P(s_t = s|s_0, a_0; \pi_1) - \sum_{s_0} \rho(s_0) \sum_{a_0} \pi_2(a_0|s_0) P(s_t = s|s_0, a_0; \pi_2)| \\
&\leq |\sum_{s_0} \rho(s_0) \sum_{a_0} \pi_1(a_0|s_0) P(s_t = s|s_0, a_0; \pi_1) - \sum_{s_0} \rho(s_0) \sum_{a_0} \pi_1(a_0|s_0) P(s_t = s|s_0, a_0; \pi_2)| \\
&\quad + |\sum_{s_0} \rho(s_0) \sum_{a_0} \pi_1(a_0|s_0) P(s_t = s|s_0, a_0; \pi_2) - \sum_{s_0} \rho(s_0) \sum_{a_0} \pi_2(a_0|s_0) P(s_t = s|s_0, a_0; \pi_2)| \\
&\leq |\sum_{s_1} d_{\pi_1}^1(s_1) \left( P(s_t = s|s_1; \pi_1) - P(s_t = s|s_1; \pi_2) \right)| + \mathbb{E}_{s_0 \sim \rho} \sum_{a_0} |\pi_1(a_0|s_0) - \pi_2(a_0|s_0)| P(s_t = s|s_0, a_0; \pi_2)
\end{aligned}
\tag{16}
$$

Add $\sum_s$ on both sides of the above equality, we get the following inequality:

$$
\begin{aligned}
&\sum_s |d_{\pi_1}^t(s) - d_{\pi_2}^t(s)| \\
&\leq \mathbb{E}_{s_1 \sim d_{\pi_1}^1} \sum_s |P(s_t = s|s_1; \pi_1) - P(s_t = s|s_1; \pi_2)| + \mathbb{E}_{s_0 \sim \rho} \|\pi_1(\cdot|s_0) - \pi_2(\cdot|s_0)\|_1 \tag{17}
\end{aligned}
$$

We can apply similar operations on $P(s_t = s|s_1; \pi_1) - P(s_t = s|s_1; \pi_2)$ as follows:

$$\mathbb{E}_{s_1 \sim d_{\pi_1}^1} \sum_s |P(s_t = s|s_1; \pi_1) - P(s_t = s|s_1; \pi_2)|$$

$$= \mathbb{E}_{s \sim d_{\pi_1}^1} \sum_s |\sum_{a_1} [\pi_1(a_1|s_1)P(s_t = s|s_1, a_1; \pi_1) - \pi_2(a_1|s_1)P(s_t = s|s_1, a_2; \pi_2)]|$$

$$\leq \mathbb{E}_{s_2 \sim d_{\pi_1}^2} \sum_s |P(s_t = s|s_2; \pi_1) - P(s_t = s|s_2; \pi_2)| + \mathbb{E}_{s_1 \sim d_{\pi_1}^1} [\|\pi_1(\cdot|s_1) - \pi_2(\cdot|s_1)\|_1]$$

Again, if we continue expand $P(s_t = s|s_2; \pi_1) - P(s_t = s|s_2; \pi_2)$ till time step $t$, we get:

$$\sum_s |d_{\pi_1}^t(s) - d_{\pi_2}^t(s)| \leq \sum_{i=0}^{t-1} \mathbb{E}_{s_i \sim d_{\pi_1}^i} [\|\pi_1(\cdot|s_i) - \pi_2(\cdot|s_i)\|_1] \tag{18}$$

Hence, for $\|d_{\pi_1} - d_{\pi_2}\|_1$, we have:

$$\|d_{\pi_1} - d_{\pi_2}\|_1 \leq (1 - \gamma) \sum_{t=0}^{\infty} \gamma^t \|d_{\pi_1}^t - d_{\pi_2}^t\|_1$$

$$\leq \sum_{t=0}^{\infty} \gamma^t \mathbb{E}_{s \sim d_{\pi_1}^t} [\|\pi_1(\cdot|s) - \pi_2(\cdot|s)\|_1] \leq \sum_{t=0}^{\infty} 2\gamma^t \mathbb{E}_{s \sim d_{\pi_1}^t} [D_{TV}(\pi_1(\cdot|s), \pi_2(\cdot|s))] \leq \frac{2\alpha}{1 - \gamma}. \tag{19}$$

$\square$

**Lemma A.2** *For any two distribution $P$ and $Q$ over $\mathcal{X}$, and any bounded function $f : \mathcal{X} \to \mathbb{R}$ such that $|f(x)| \leq c, \forall x \in \mathcal{X}$, we have:*

$$|\mathbb{E}_{x \sim P}[f(x)] - \mathbb{E}_{x \sim Q}[f(x)]| \leq c\|P - Q\|_1. \tag{20}$$

**Proof:**

$$|\mathbb{E}_{x \sim P}[f(x)] - \mathbb{E}_{x \sim Q}[f(x)]| = |\sum_{x \in \mathcal{X}} P(x)f(x) - Q(x)f(x)|$$

$$\leq \sum_x |P(x)f(x) - Q(x)f(x)| \leq \sum_x |f(x)||P(x) - Q(x)|$$

$$\leq c \sum_x |P(x) - Q(x)| = c\|P - Q\|_1. \tag{21}$$

$\square$

## A.2 Proof of Theorem 3.1

Recall that we denote $d_\pi \pi$ as the joint state-action distribution under policy $\pi$. To prove Theorem 3.1, we will use Lemma 1.2 presented in the Appendix from [37] to prove the following claim:

**Lemma A.3** *Suppose we learned a approximate model $\hat{P}$ and obtain the optimal policy $\eta_n$ with respect to the objective function $J(\pi)$ under $\hat{P}$ and the trust-region constraint $\mathbb{E}_{s \sim d_{\pi_n}} D_{TV}(\pi, \pi_n) \leq \alpha$, then compare to $\pi_n^*$, we have:*

$$J(\eta_n) - J(\eta_n^*) \leq \frac{\gamma}{2(1-\gamma)} \left( \mathbb{E}_{(s,a) \sim d_{\eta_n} \eta_n} \left[ \|\hat{P}_{s,a} - P_{s,a}\|_1 \right] + \mathbb{E}_{(s,a) \sim d_{\eta_n^*} \eta_n^*} \left[ \|\hat{P}_{s,a} - P_{s,a}\|_1 \right] \right). \tag{22}$$

**Proof:** Denote $\hat{V}^\pi$ as the value function of policy $\pi$ under the approximate model $\hat{P}$. From Lemma 1.2 and Corollary 1.2 in [37], we know that for any two policies $\pi_1$ and $\pi_2$, we have:

$$J(\pi_1) - J(\pi_2) = \mathbb{E}_{s \sim \rho_0}[\hat{V}^{\pi_1}(s) - \hat{V}^{\pi_2}(s)]$$

$$+ \frac{\gamma}{2(1-\gamma)} \left( \mathbb{E}_{(s,a) \sim d_{\pi_1} \pi_1} \left[ \|\hat{P}_{s,a} - P_{s,a}\|_1 \right] + \mathbb{E}_{(s,a) \sim d_{\pi_2} \pi_2} \left[ \|\hat{P}_{s,a} - P_{s,a}\|_1 \right] \right). \tag{23}$$

Now replace $\pi_1$ with $\eta_n$ and $\pi_2$ with $\eta_n^*$. Note that both $\eta_n$ and $\eta_n^*$ are in the trust region constraint $\mathbb{E}_{s \sim d_{\pi_n}} D_{TV}(\pi, \pi_n) \leq \alpha$ by definition. As $\eta_n$ is the optimal control under the approximate model $\hat{P}$ (i.e., the optimal solution to Eq. 4), we must have $\mathbb{E}_{s \sim \rho_0}[\hat{V}^{\eta_n}(s) - \hat{V}^{\eta_n^*}(s)] \leq 0$. Substitute it back to Eq. 23, we immediately prove the above lemma. $\qquad\square$

The above lemma shows that the performance gap between $\eta_n$ and $\eta_n^*$ is measured under the state-action distributions measured from $\eta_n$ and $\eta_n^*$ while our model $\hat{P}$ is only accurate under the state-action distribution from $\pi_n$. Luckily due to the trust-region constraint $\mathbb{E}_{s \sim d_{\pi_n}} D_{TV}(\pi, \pi_n)$ and the fact that $\eta_n$ and $\eta_n^*$ are both in the trust-region, we can show that $d_{\eta_n} \eta_n$, $d_{\pi_n^*} \pi_n^*$ are not that far from $d_{\pi_n} \pi_n$ using Lemma A.1:

$$\|d_{\eta_n} \eta_n - d_{\pi_n} \pi_n\|_1 \leq \|d_{\eta_n} \eta_n - d_{\pi_n} \eta_n\|_1 + \|d_{\pi_n} \eta_n - d_{\pi_n} \pi_n\|_1$$

$$\leq \|d_{\eta_n} - d_{\pi_n}\|_1 + \mathbb{E}_{s \sim d_{\pi_n}}[\|\eta_n(\cdot|s) - \pi_n(\cdot|s)\|_1] \leq \frac{2\alpha}{1 - \gamma} + 2\alpha \leq \frac{4\alpha}{1 - \gamma}. \tag{24}$$

similarly, for $\pi_n^*$ we have:

$$\|d_{\eta_n^*} \eta_n^* - d_{\pi_n} \pi_n\|_1 \leq \frac{4\alpha}{1 - \gamma}. \tag{25}$$

Go back to Eq. 22, let us replace $\mathbb{E}_{d_{\eta_n} \eta_n}$ and $\mathbb{E}_{d_{\eta_n^*} \eta_n^*}$ by $\mathbb{E}_{d_{\pi_n} \pi_n}$ and using Lemma A.2, we will have:

$$|\mathbb{E}_{(s,a) \sim d_{\eta_n} \eta_n}[\|\hat{P}_{s,a} - P_{s,a}\|_1] - \mathbb{E}_{(s,a) \sim d_{\pi_n} \pi_n}[\|\hat{P}_{s,a} - P_{s,a}\|_1]| \leq 2\|d_{\eta_n} \eta_n - d_{\pi_n} \pi_n\|_1 \leq \frac{8\alpha}{1 - \gamma}$$

$$\Rightarrow \mathbb{E}_{(s,a) \sim d_{\eta_n} \eta_n}[\|\hat{P}_{s,a} - P_{s,a}\|_1] \leq \mathbb{E}_{(s,a) \sim d_{\pi_n} \pi_n}[\|\hat{P}_{s,a} - P_{s,a}\|_1] + \frac{8\alpha}{(1 - \gamma)}, \tag{26}$$

and similarly,

$$\mathbb{E}_{(s,a) \sim d_{\eta_n^*} \eta_n^*}[\|\hat{P}_{s,a} - P_{s,a}\|_1] \leq \mathbb{E}_{(s,a) \sim d_{\pi_n} \pi_n}[\|\hat{P}_{s,a} - P_{s,a}\|_1] + \frac{8\alpha}{(1 - \gamma)}. \tag{27}$$

Combine Eqs. 26 and 27, we have:

$$J(\eta_n) - J(\eta_n^*) \leq \frac{\gamma}{2(1 - \gamma)} \left( 2\mathbb{E}_{(s,a) \sim d_{\pi_n} \pi_n}[\|\hat{P}_{s,a} - P_{s,a}\|_1] + 16\alpha/(1 - \gamma) \right)$$

$$= \frac{\gamma\delta}{1 - \gamma} + \frac{8\gamma\alpha}{(1 - \gamma)^2} = O\left(\frac{\gamma\delta}{1 - \gamma}\right) + O\left(\frac{\gamma\alpha}{(1 - \gamma)^2}\right). \tag{28}$$

Using the definition of $\Delta(\alpha)$, adding $J(\pi_n)$ and subtracting $J(\pi_n)$ on the LHS of the above inequality, we prove the theorem.

### A.3 Proof of Theorem 4.1

The definition of $\pi_{n+1}$ implies that $\mathbb{E}_{s \sim d_{\pi_n}}[D_{TV}(\pi_{n+1}(\cdot|s), \pi_n(\cdot|s))] \leq \beta$. Using Lemma A.1, we will have that the total variation distance between $d_{\pi_{n+1}}^t$ and $d_{\pi_n}^t$ is:

$$\|d_{\pi_{n+1}} - d_{\pi_n}\|_1 \leq \frac{2\beta}{1 - \gamma}. \tag{29}$$

Now we can compute the performance improvement of $\pi_{n+1}$ over $\eta_n$ as follows:

$$(1 - \gamma)(J(\pi_{n+1}) - J(\eta_n)) = \mathbb{E}_{s \sim d_{\pi_{n+1}}}\left[\mathbb{E}_{a \sim \pi_{n+1}}[A^{\eta_n}(s, a)]\right]$$

$$= \mathbb{E}_{s \sim d_{\pi_{n+1}}}\left[\mathbb{E}_{a \sim \pi_{n+1}}[A^{\eta_n}(s, a)]\right] - \mathbb{E}_{s \sim d_{\pi_n}}\left[\mathbb{E}_{a \sim \pi_{n+1}}[A^{\eta_n}(s, a)]\right] + \mathbb{E}_{s \sim d_{\pi_n}}\left[\mathbb{E}_{a \sim \pi_{n+1}}[A^{\eta_n}(s, a)]\right]$$

$$\leq \left|\mathbb{E}_{s \sim d_{\pi_{n+1}}}\left[\mathbb{E}_{a \sim \pi_{n+1}}[A^{\eta_n}(s, a)]\right] - \mathbb{E}_{s \sim d_{\pi_n}}\left[\mathbb{E}_{a \sim \pi_{n+1}}[A^{\eta_n}(s, a)]\right]\right| + \mathbb{E}_{s \sim d_{\pi_n}}\left[\mathbb{E}_{a \sim \pi_{n+1}}[A^{\eta_n}(s, a)]\right]$$

$$\leq \frac{2\varepsilon\beta}{1 - \gamma} + \mathbb{E}_{s \sim d_{\pi_n}}\left[\mathbb{E}_{a \sim \pi_{n+1}}[A^{\eta_n}(s, a)]\right]$$

$$= \frac{2\varepsilon\beta}{1 - \gamma} + \mathbb{A}_n(\pi_{n+1})$$

$$= \frac{2\varepsilon\beta}{1 - \gamma} - |\mathbb{A}_n(\pi_{n+1})| \tag{30}$$

Finally, to bound $J(\pi_{n+1}) - J(\pi_n)$, we can simply do:

$$J(\pi_{n+1}) - J(\pi_n) = J(\pi_{n+1}) - J(\eta_n) + J(\eta_n) - J(\pi_n)$$

$$\leq \frac{\beta\varepsilon}{(1-\gamma)^2} - \frac{|\mathbb{A}_n(\pi_{n+1})|}{1-\gamma} - \Delta(\alpha). \tag{31}$$

## A.4 Global Performance Guarantee for DPI

When $|\Delta_n(\alpha)|$ and $|\mathbb{A}_n(\pi_{n+1})|$ are small, say $|\Delta_n(\alpha)| \leq \xi, |\mathbb{A}_n(\pi_{n+1})| \leq \xi$, then we can guarantee that $\eta_n$ and $\pi_n$ are good policies, if our initial distribution $\rho$ happens to be a good distribution (e.g., close to $d_{\pi^*}$), and the *realizable assumption* holds: $\min_{\pi \in \Pi} \mathbb{E}_{s \sim d_{\pi_n}}\left[\mathbb{E}_{a \sim \pi(\cdot|s)}[A^{\eta_n}(s,a)]\right] = \mathbb{E}_{s \sim d_{\pi_n}}\left[\min_{a \sim \mathcal{A}}[A^{\eta_n}(s,a)]\right]$. We call a policy class $\Pi$ *closed under its convex hull* if for any sequence of policies $\{\pi_i\}_i, \pi_i \in \Pi$, the convex combination $\sum_i w_i \pi_i$, for any $w$ such that $w_i \geq 0$ and $\sum_i w_i = 1$, also belongs to $\Pi$.

**Theorem A.4** *Under the realizable assumption and the assumption of $\Pi$ is closed under its convex hull, and $\max\{|\mathbb{A}_n(\pi_{n+1})|, \Delta(\alpha)\} \leq \xi \in \mathbb{R}^+$, then for $\eta_n$, we have:*

$$J(\eta_n) - J(\pi^*) \leq \left(\max_s \left(\frac{d_{\pi^*}(s)}{\rho_0(s)}\right)\right) \left(\frac{\xi}{\beta(1-\gamma)^2} + \frac{\xi}{\beta(1-\gamma)}\right).$$

The term $(\max_s (d_{\pi^*}(s)/\rho_0(s)))$ measures the distribution mismatch between the initial state distribution $\rho_0$ and the optimal policy $\pi^*$, and appears in some of previous API algorithms–CPI [5] and PSDP [4].[4]

**Proof:** Recall the average advantage of $\pi_{n+1}$ over $\pi_n$ is defined as $\mathbb{A}_{\pi_n}(\pi_{n+1}) = \mathbb{E}_{s \sim d_{\pi_n}}\left[\mathbb{E}_{a \sim \pi_{n+1}(\cdot|s)}[A^{\eta_n}(s,a)]\right]$. Also recall that the conservative update where we first compute $\pi_n^* = \arg\min_{\pi \in \Pi} \mathbb{E}_{s \sim d_{\pi_n}}[\mathbb{E}_{a \sim \pi} A^{\eta_n}(s,a)]$, and then compute the new policy $\pi'_{n+1} = (1-\beta)\pi_n + \beta\pi_n^*$. Note that under the assumption that the policy class $\Pi$ is closed under its convex hull, we have that $\pi'_{n+1} \in \Pi$. As we showed that $\pi'_{n+1}$ satisfies the trust-region constraint defined in Eq. 5, we must have:

$$\mathbb{A}_{\pi_n}(\pi_{n+1}) = \mathbb{E}_{s \sim d_{\pi_n}}\left[\mathbb{E}_{a \sim \pi_{n+1}(\cdot|s)}[A^{\eta_n}(s,a)]\right] \leq \mathbb{E}_{s \sim d_{\pi_n}}\left[\mathbb{E}_{s \sim \pi'_{n+1}}[A^{\eta_n}(s,a)]\right], \tag{32}$$

due to the fact that $\pi_{n+1}$ is the optimal solution of the optimization problem shown in Eq. 5 subject to the trust region constraint. Hence if $\mathbb{A}_\pi(\pi_{n+1}) \geq -\xi$, we must have $\mathbb{E}_{s \sim d_{\pi_n}}\left[\mathbb{E}_{s \sim \pi'_{n+1}}[A^{\eta_n}(s,a)]\right] \geq -\xi$, which means that:

$$\mathbb{E}_{s \sim d_{\pi_n}}\left[(1-\beta)\mathbb{E}_{s \sim d_{\pi_n}} A^{\eta_n}(s,a) + \beta\mathbb{E}_{s \sim d_{\pi_n^*}} A^{\eta_n}(s,a)\right]$$

$$= (1-\beta)(1-\gamma)(J(\pi_n) - J(\eta_n)) + \beta\mathbb{E}_{s \sim d_{\pi_n}}[\mathbb{E}_{a \sim \pi_n^*} A^{\eta_n}(s,a)] \geq -\xi,$$

$$\Rightarrow \mathbb{E}_{s \sim d_{\pi_n}}[\mathbb{E}_{a \sim \pi_n^*} A^{\eta_n}(s,a)] \geq -\frac{\xi}{\beta} - \frac{1-\beta}{\beta}(1-\gamma)\Delta(\alpha) \geq -\frac{\xi}{\beta} - \frac{1-\gamma}{\beta}\Delta(\alpha). \tag{33}$$

Recall the realizable assumption: $\mathbb{E}_{s \sim d_{\pi_n}}[\mathbb{E}_{a \sim \pi_n^*} A^{\eta_n}(s,a)] = \mathbb{E}_{s \sim d_{\pi_n}}[\min_a A^{\eta_n}(s,a)]$, we have:

$$-\frac{\xi}{\beta} - \frac{1-\gamma}{\beta}\Delta(\alpha) \leq \sum_s d_{\pi_n}(s)\min_a A^{\eta_n}(s,a) = \sum_s \frac{d_{\pi_n}(s)}{d_{\pi^*}(s)}d_{\pi^*}(s)\min_a A^{\eta_n}(s,a)$$

$$\leq \min_s\left(\frac{d_{\pi_n}(s)}{d_{\pi^*}(s)}\right)\sum_s d_{\pi^*}(s)\min_a A^{\eta_n}(s,a)$$

$$\leq \min_s\left(\frac{d_{\pi_n}(s)}{d_{\pi^*}(s)}\right)\sum_s d_{\pi^*}(s)\sum_a \pi^*(a|s)A^{\eta_n}(s,a)$$

$$= \min_s\left(\frac{d_{\pi_n}(s)}{d_{\pi^*}(s)}\right)(1-\gamma)(J(\pi^*) - J(\eta_n)). \tag{34}$$

Rearrange, we get:

$$J(\eta_n) - J(\pi^*) \leq \left(\max_s \left(\frac{d_{\pi^*}(s)}{d_{\pi_n}(s)}\right)\right)\left(\frac{\xi}{\beta(1-\gamma)} + \frac{\Delta(\alpha)}{\beta}\right)$$

$$\leq \left(\max_s \left(\frac{d_{\pi^*}(s)}{\rho(s)}\right)\right)\left(\frac{\xi}{\beta(1-\gamma)^2} + \frac{\xi}{\beta(1-\gamma)}\right) \tag{35}$$

$\square$

## A.5 Analysis on Using DAGGER for Updating $\pi_n$

Note that in ExIt, once an intermediate expert is constructed, DAGGER [38] is used as the imitation learning algorithm to improvement the reactive policy. DAGGER does not directly optimize the ultimate objective function—the expected total cost, but instead trying minimizes the number of mismatches between the learner and the expert. Here we used more advanced, cost-aware IL algorithms, AGGREVATE [26] and AGGREVATED [27], which directly reason about the expected total cost, and guarantee to learn a policy that achieves one-step deviation improvement of the expert policy.

Below we analyze the update of $\pi$ using DAGGER.

To analyze the update of $\pi$ using DAGGER, we consider deterministic policy here: we assume $\pi_n$ and $\eta$ are both deterministic and the action space $\mathcal{A}$ is discrete. We consider the following update procedure for $\pi$:

$$\min_{\pi \in \Pi} \mathbb{E}_{s \sim d_{\pi_n}}\left[\mathbb{E}_{a \sim \pi(\cdot|s)}\mathbb{1}(a \neq \arg\min_a A^{\eta_n}(s,a))\right],$$

$$s.t., \mathbb{E}_{s \sim d_{\pi_n}}[\|\pi(\cdot|s) - \pi_n(\cdot|s)\|_1] \leq \beta. \tag{36}$$

Namely we simply convert the cost vector defined by the disadvantage function by a "one-hot" encoded cost vector, where all entries are 1, except the entry corresponding to $\arg\min_a A^{\eta_n}(s,a)$ has cost 0. Ignoring the updates on the "expert" $\eta_n$, running the above update step with respect to $\pi$ can be regarded as running online gradient descent with a local metric defined by the trust-region constraint. Recall that $\eta_n$ may from a different policy class than $\Pi$.

Assume that we learn a policy $\pi_{n+1}$ that achieves $\epsilon_n$ prediction error:

$$\mathbb{E}_{s \sim d_{\pi_n}}\left[\mathbb{E}_{a \sim \pi_{n+1}(\cdot|s)}\left[\mathbb{1}(a \neq \arg\min_a A^{\eta_n}(s,a))\right]\right] \leq \epsilon_n. \tag{37}$$

Namely we assume that we learn a policy $\pi_{n+1}$ such that the average probability of mismatch to $\eta_n$ is at most $\epsilon_n$.

Using Lemma A.1, we will have that the total variation distance between $d_{\pi_{n+1}}$ and $d_{\pi_n}$ is at most:

$$\|d_{\pi_{n+1}} - d_{\pi_n}\|_1 \leq \frac{2\beta}{1-\gamma}. \tag{38}$$

Applying PDL, we have:

$$(1-\gamma)(J(\pi_{n+1}) - J(\eta_n)) = \mathbb{E}_{s \sim d_{\pi_{n+1}}}[\mathbb{E}_{a \sim \pi_{n+1}}[A^{\eta_n}(s,a)]]$$

$$\leq \mathbb{E}_{s \sim d_{\pi_n}}[\mathbb{E}_{a \sim \pi_{n+1}}[A^{\eta_n}(s,a)]] + \frac{2\beta\varepsilon}{1-\gamma}$$

$$= \mathbb{E}_{s \sim d_{\pi_n}}\left[\sum_{a \neq \arg\min_a A^{\eta_n}(s,a)} \pi(a|s)A^{\eta_n}(s,a)\right] + \frac{2\beta\varepsilon}{1-\gamma}$$

$$\leq (\max_{s,a} A^{\eta_n}(s,a))\mathbb{E}_{s \sim d_{\pi_n}}[\mathbb{E}_{a \sim \pi_{n+1}}\mathbb{1}(a \neq \arg\min_a A^{\eta_n}(s,a))] + \frac{2\beta\varepsilon}{1-\gamma}$$

$$\leq \varepsilon'\epsilon_n + \frac{2\beta\varepsilon}{1-\gamma}, \tag{39}$$

where we define $\varepsilon' = \max_{s,a} A^{\eta_n}(s,a)$, which should be at a similar scale as $\varepsilon$. Hence we can show that performance difference between $\pi_{n+1}$ and $\pi_n$ as:

$$J(\pi_{n+1}) - J(\pi_n) \leq \frac{2\beta\epsilon}{(1-\gamma)^2} + \frac{\varepsilon'\epsilon_n}{1-\gamma} - \Delta(\alpha). \tag{40}$$

Now we can compare the above upper bound to the upper bound shown in Theorem 4.1. Note that even if we assume the policy class is rich and the learning process perfect learns a policy (i.e., $\pi_{n+1} = \eta_n$) that achieves prediction error $\epsilon_n = 0$, we can see that the improvement of $\pi_{n+1}$ over $\pi_n$ only consists of the improvement from the local optimal control $\Delta(\alpha)$. While in theorem 4.1, under the same assumption, except for $\Delta(\alpha)$, the improvement of $\pi_{n+1}$ over $\pi_n$ has an extra term $\frac{|\mathbb{A}_n(\pi_{n+1})|}{1-\gamma}$, which basically indicates that we learn a policy $\pi_{n+1}$ that is one-step deviation improved over $\eta_n$ by leveraging the cost informed by the disadvantage function. If one uses DAgger, than the best we can hope is to learn a policy that performs as good as the "expert" $\eta_n$ (i.e., $\epsilon_n = 0$).

# B  Additional Experimental Details

## B.1  Synthetic Discrete MDPs and Conservative Policy Update Implementation

We follow [7] to randomly create 10 discrete MDPs, each with 1000 states, 5 actions and 2 branches (namely, each state action pair leads to at most 2 different states in the next step). We work in model-free setting: we cannot explicitly compute the distribution $d_\pi$ and we can only generate samples from $d_\pi$ by executing $\pi$.

We maintain a tabular representation $\hat{P} \in \mathbb{R}^{|\mathcal{S}| \times |\mathcal{A}| \times |\mathcal{S}|}$, where each entry $P_{i,j,k}$ records the number of visits of the state-action-next state triple. We represent $\eta$ as a 2d matrix $\eta \in \mathbb{R}^{|\mathcal{S} \times \mathcal{A}|}$, where $\eta_{i,j}$ stands for the probability of executing action $j$ at state $i$. The reactive policy uses the binary encoding of the state id as the feature, which we denote as $\phi(s) \in \mathbb{R}^{d_s}$ ($d_s$ is the dimension of feature space, which is $\log_2(|\mathcal{S}|)$ in our setup). Hence the reactive policy $\pi_n$ sits in low-dimensional feature space and doesn't scale with respect to the size of the state space $S$.

For both our approach and CPI, we implement the unconstrained cost-sensitive classification (Eq. 5) by the Cost-Sensitive One Against All (CSOAA) classification technique. Specifically, given a set of states $\{s_i\}_i$ sampled from $d_{\pi_n}$, and a cost vector $\{A^{\eta_n}(s_i, \cdot) \in \mathbb{R}^{|\mathcal{A}|}\}$ (byproduct of VI), we train a linear regressor $\hat{W} \in \mathbb{R}^{|\mathcal{A}| \times d_s}$ to predict the cost vector: $\hat{W}\phi(s) \approx \mathcal{A}^{\eta_n}(s, \cdot)$. Then $\pi_n^*$ in Eq. 6 is just a classifier that predicts action $\arg\min_i(\hat{W}s)[i]$ corresponding to the smallest predicted cost. We then combine $\pi_n^*$ with the previous policies as shown in Eq. 6 to make sure $\pi_{n+1}$ satisfies the trust region constraint in Eq. 5.

For CPI, we estimate $A^{\pi_n}(s, a)$ by running value iteration using $\hat{P}$ with the original cost matrix. We also experimented estimating $A^{\pi_n}(s, \cdot)$ by empirical rollouts with importance weighting, which did not work well in practice due to high variance resulting from the empirical estimate and importance weight. For our method, we alternately compute $\eta_n$ using VI with the new cost shown in Eq. 10 and $\hat{P}$, and update the Lagrange multiplier $\mu$, under convergence. Hence *the only difference* between our approach and CPI here is simply that we use $A^{\eta_n}$ while CPI uses $A^{\pi_n}$. We tuned the step size $\beta$ (Eq. 6) for CPI. Neither our method nor CPI used line-search trick for $\beta$.

Our results indicates that using $A^{\eta_n}$ converges much faster than using $A^{\pi_n}$, although computing $\eta_n$ is much more time consuming than computing $A^{\pi_n}$. But again we emphasize that computing $\eta_n$ doesn't require extra samples. For real large discrete MDPs, we can easily plug in an approximate VI techniques such as [39] to significantly speed up computing $\eta_n$.

## B.2  Details for Updating Lagrange Multiplier $\mu$

Though running gradient ascent on $\mu$ is theoretically sound and can work in practice as well, but it converges slow and requires to tune the learning rate as we found experimentally. To speed up convergence, we used the same update procedure used in the practical implementation of Guided Policy Search [40]. We set up $\mu_{\min}$ and $\mu_{\max}$. Starting from $\mu = \mu_{\min}$, we fix $\mu$ and compute $\eta$ using the new cost $c'$ as shown in Eq. 10 under the local dynamics $\hat{P}$ using LQR. We then compare $\mathbb{E}_{s \sim \mu_n} D_{KL}(\eta(\cdot|s), \pi_n(\cdot|s))$ to $\alpha$. If $\eta$ violates the constraint, i.e., $\mathbb{E}_{s \sim \mu_n} D_{KL}(\eta(\cdot|s), \pi_n(\cdot|s)) > \alpha$, then it means that $\mu$ is too small. In this case, we set $\mu_{\min} = \mu$, and compute new $\mu$ as $\mu = \min(\sqrt{\mu_{\min}\mu_{\max}}, 10\mu_{\min})$; On the other hand, if $\eta$ satisfies the KL constraint, i.e, $\mu$ is too big, we set $\mu_{\max} = \mu$, and compute new $\mu$ as $\mu = \max(\sqrt{\mu_{\min}\mu_{\max}}, 0.1\mu_{\max})$. We early terminate the process once we find $\eta$ such that $0.9\alpha \leq \mathbb{E}_{s \sim \mu_n} D_{KL}(\eta(\cdot|s), \pi_n(\cdot|s)) \leq 1.1\alpha$. We then store the most recent Lagrange multiplier $\mu$ which will be used as warm start of $\mu$ for the next iteration.

## B.3 Details on the Natural Gradient Update to $\pi$

Here we provide details for updating $\pi_\theta$ via imitating $\eta_\Theta$ (Sec. 5.2) Recall the objective function $L_n(\theta) = \mathbb{E}_{s \sim d_{\pi_{\theta_n}}} [\mathbb{E}_{a \sim \pi(\cdot|s;\theta)}[A^{\eta_{\Theta_n}}(s,a)]]$, $\nabla_{\theta_n}$ as $\nabla_\theta L_n(\theta)|_{\theta=\theta_n}$, and $F_{\theta_n}$ is the Fisher information matrix (equal to the Hessian of the KL constraint measured at $\theta_n$). To compute $F_\theta^{-1} \nabla_\theta$, in our implementation, we use Conjugate Gradient with the Hessian-vector product trick [12] to directly compute $F^{-1}\nabla$.

Note that the unbiased empirical estimation of $\nabla_{\theta_n}$ and $F_{\theta_n}$ is well-studied and can be computed using samples generated from executing $\pi_{\theta_n}$. Assume we roll out $\pi_{\theta_n}$ to generate $K$ trajectories $\tau^i = \{s_0^i, a_0^i, ...s_T^i, a_T^i\}, \forall i \in [K]$. The empirical gradient and Fisher matrix can be formed using these samples as $\nabla_{\theta_n} = \sum_{s,a} [\nabla_{\theta_n}(\ln(\pi(a|s;\theta_n))) A^{\eta_{\Theta_n}}(s,a)]$ and $F_{\theta_n} = \sum_{s,a} [(\nabla \ln(\pi(a|s;\theta_n)))(\nabla_{\theta_n} \ln(\pi(a|s;\theta_n))^T]$.

The objective $L_n(\theta)$ could be nonlinear with respect $\theta$, depending on the function approximator used for $\pi_n,$. Hence one step of gradient descent may not reduce $L_n(\theta)$ enough. In practice, we can perform $k$ steps $(k > 1)$ of NGD shown in Eq. 12, with the learning rate shrinking to $\sqrt{(\beta/k)/(\nabla_\theta^T F_{\theta_n}^{-1} \nabla_\theta)}$ to ensure that after $k$ steps, the solution still satisfies the constraint in Eq. 11.

## B.4 Details on the Continuous Control Experiment Setup

The cost function $c(s,a)$ for discrete MDP is uniformly sampled from $[0,1]$. For the continuous control experiments, we designed the cost function $c(s,a)$, which is set to be known to our algorithms. For cartpole and helicopter hover, denote the target state as $s^*$, the cost function is designed to be exactly quadratic: $c(s,a) = (s - s^*)^T Q(s - s^*) + a^T R a$, which penalizes the distance to the goal and large control inputs. For Swimmer, Hopper and Half-Cheetah experiment, we set up a target moving forward speed $v^*$. For any state, denote the velocity component as $s_v$, the quadratic cost function is designed as $c(s,a) = q(s_v - v^*)^2 + a^T R a$, which encourages the agent to move forward in a constant speed while avoiding using large control inputs.

For reactive policies, we simply used two-layer feedforward neural network as the parameterized policies–the same structures used in the implementation of [12].

For local linear model fit under $d_\pi$, given a dataset in the format of $\{s_i, a_i, s_i'\}_{i=1}^N$, where $s_i \sim d_\pi$, $a_i \sim \pi(\cdot|s)$, and $s_i' \sim P_{s_i,a_i}$, we perform Ridge linear regression:

$$A, B, c = \arg \min_{A,B,c} \frac{1}{N} \sum_{i=1}^N \|As_i + Ba_i + c - s_i'\|_2^2 + \lambda(\|A\|_F + \|B\|_F + \|c\|_2), \quad (41)$$

where regularization $\lambda$ is pre-fixed to be a small number for all experiments. With $A, B, c$, we estimate $\Sigma$ as $\Sigma = \frac{1}{N} \sum_{i=1}^N e_i e_i^T$, where $e_i = As_i + Ba_i + c - s_i'$.

## Footnotes

[4]While CPI considers a different setting where a good reset distribution $\nu$ (different from $\rho_0$) is accessible, DPI can utilize such reset distribution by replacing $\rho_0$ by $\nu$ during training.