[Reviews · NeurIPS 2018]

Reviewer 1



Major Comments: 1) page 3: How do you set a parameter $\alpha$ and what effect it has? In other words, if $\eta^* = \argmin J(\eta)$ and $\eta_{\alpha}$ is the solution of Eq , what can you say about relation between $\eta^*$ and $\eta_{\alpha}$? 2) page 3: parameter $\delta$ in Eq 3. Could you please elaborate more on how are you going to choose? If it does depend on policy $\pi_n$, then we need to replace it by $\delta_n$ (i.e. it is changing from iteration to iteration). Also, apparently, this parameter will have an effect on the transition policy $\hat{P}$ that is found, and, hence, has effect to the policy $\eta_n$ evaluated at iteration $n$. Did you study this effect? 3) page 3: Theorem 3.1. I am confused with this result. Because the real transition $P$ is not available then we need to study the $J(\eta_n)$ evaluated with model $\hat{P}$ instead of $P$. But Theorem 3.1 studies the quality of policy $\eta_n$ with respect to real model $P$. Therefore, Theorem 3.1 doesn't quantify the quality of the tree search policy $\eta_n$. 4) page 4: Section 3.2. How do you define parameter $\beta$ in Eq 6. How effect does it have on the approximate solution $\pi_{n+1}$? 5) Due to PDL Lemma, for a given fast reactive policy $\pi_n$ the corresponding tree search policy $\eta_n$ actually doesn't depend on $\pi_n$. This is because policy $eta_n = argmin_{\eta} J(\eta)$, and $J(\eta)$ doesn’t depend on $pi_n$. The dependence appears when we impose trust region constraint given in terms of $\pi_n$. My biggest issue with this paper is about the parameters $\alpha, \delta, \beta$ that was introduced as some form of relaxation but, at the same time, no effect of them were studied. (Please see my comment on Theorem 3.1 in 3) for better understanding). Minor Comments: 1) page 3: Although the min-max framework considered in Section 3 is understandable, it would be good to elaborate on for readers. Apart from the above major and minor comments, overall the paper appears to be written and structured well. Technically novel.

Reviewer 2



This paper studies a concept called dual policy iteration, a bunch of extensions from a theoretical perspective is provided. Albeit that several examples are provided to show the superiority of the proposed method, the evaluations are still not convincing. I would like to see if other two-policy strategy approaches (such as those mentioned in the second paragraph in ‘introduction’) can perform well, without that I can not judge the practical contribution of the presented work.

Reviewer 3



It is a good work because the authors have proposed a new optimization problem for approximate policy iteration methods and achieved a better policy improvement than the conservative policy iteration. However, there is one problem confusing me. In the practical instance of DPI, LQR works because cost function is designed to be approximated so that the approximation error would be low. But the cost function would not always be so smooth. In other words, the practical instance of DPI may fail if the cost function could not be approximated by quadratic function. Besides, why \eta is designed to be time-varying while \pi is a single policy as these two are both policies. And please show the scores of the continuous settings. Although the experiments of continuous settings shows a better performance of the proposed algorithm than TRPO-GAE under the cumulative cost defined in the paper, it lacks the average scores comparison of these experiments by default reward settings.